# The Role of Organizational Factors in Nurses’ Perceived Preparedness to Screen, Intervene and Refer in Cases of Suspected Postpartum Depression

**DOI:** 10.3390/ijerph192416717

**Published:** 2022-12-13

**Authors:** Rena Bina, Saralee Glasser, Mira Honovich, Yona Ferber, Samira Alfayumi-Zeadna

**Affiliations:** 1School of Social Work, Bar Ilan University, Webb St., Ramat Gan 52900, Israel; 2Gertner Institute for Epidemiology & Health Policy Research, Tel Hashomer, Ramat Gan 52621, Israel; 3Public Health Nursing, Ministry of Health, Jerusalem 9446724, Israel; 4Nursing Department, School of Health Sciences, Ashkelon Academic College, Ashkelon 78682, Israel; 5Center for Women’s Health Studies and Promotion, Ben-Gurion University of the Negev, Beer Sheva 84417, Israel

**Keywords:** public health nurses, perceived preparedness, screening, postpartum depression, organization

## Abstract

Routine screening for postpartum depression (PPD) is widespread, yet little attention has been given to the perceived preparedness of health providers to perform screening procedures, or to the role of organizational factors in their preparedness, although these are crucial elements for optimal implementation. The aim of this study was to examine organizational factors associated with public health nurses’ (PHNs) perceived preparedness to screen women for PPD, intervene, and refer them in cases of suspected PPD. Two hundred and nineteen PHNs completed a self-report survey regarding their perceived preparedness to carry out a screening program (including screening, intervening, and referring women), and their perceived organizational support, supervisor’s support, colleagues’ support, and colleagues’ preparedness. A path analysis model was used to analyze the data. The results showed that perceived colleagues’ preparedness was significantly associated with the three perceived preparedness constructs (screen, intervene, and refer). Perceived supervisor’s support was positively associated with perceived preparedness to screen, and perceived organizational support was positively associated with perceived preparedness to intervene. This paper highlights the manner in which formal and informal organizational factors play an important role in the perceived preparedness of PHNs to carry out a PPD screening program, and how these factors impact the three different aspects of the screening program. Organizations implementing PPD screening should support PHNs in all aspects of the implementation process, provide guidance, and enhance peer-group continued learning through which PHNs could share knowledge, discuss barriers, and foster professional development.

## 1. Introduction

Over the past years increasing attention has been paid to the early detection of depressive symptoms in the postpartum period. As a result, in various countries around the world, screening procedures for postpartum depression (PPD) have been recommended and implemented, generally conducted by nurses [1,2]. For instance, in Israel, a mandated universal screening program has been implemented since 2013, in which public health nurses screen women for PPD symptoms at about two months postpartum, using the Edinburgh Postnatal Depression Scale (EPDS) [3]. The screenings are conducted at Mother-and-Child Healthcare Clinics (MCHC), which offer free wellness check visits and immunizations for all babies and children aged 0–5 (Israel Ministry of Health, Directive No. 20/12, 2012). Women who score 10 or above on the EPDS are offered supportive counseling by the PHNs (the “intervention” element) and/or are referred (the “referral” element) to a mental health professional for diagnosis and treatment, as needed. These screening, intervention, and referral tasks were added to the PHNs’ routine tasks following a one-to-two-day training session because they had not previously been trained specifically for dealing with emotional issues. In addition to the original training, ongoing supervision and consultation are provided by their supervisors or by a psychiatric nurse referent, and supervisors periodically conduct lectures and simulations, etc., on the topic [4].

Despite the wide-scale implementation of PPD screening procedures, emerging studies have shown that healthcare providers do not feel efficacious enough in screening women for PPD symptoms [5], and feel they are not properly prepared for carrying out such a task [6]. Little attention has been paid to healthcare providers’ experiences in conducting such procedures [7] or to their perceived preparedness to carry out a PPD screening program, although this is a crucial element for optimal implementation [8]. Their subjective perception of preparedness is an important aspect and predictor of their actual preparedness [9], of the effectiveness of care provided [10], and of the success of the intervention [11]. Moreover, organizational factors have been found to be important in influencing work performance [12]. 

Previous studies have examined the role of organizational factors in affecting PHNs’ performance of screening and intervention procedures in primary care on issues such as substance abuse, intimate partner violence, and sexual violence, however not regarding PPD. For instance, such studies have examined the success of an alcohol screening program and its association with organization-level factors [13,14]. Yet, to date, no study has been reported regarding the role of organizational factors in perceived preparedness to carry out a PPD screening program. 

The role of organizational factors in influencing workers’ behaviors has been studied extensively over the years, largely through the prism of Social Exchange Theory (SET). SET posits that a series of social interactions lead to obligations, suggesting that one person’s actions are contingent upon the actions of another [15] and that a balance in social exchanges at the workplace forms the basis for workers’ behaviors and attitudes [16]. Studies examining SET in the workplace have primarily focused on the relationship between worker and organization, termed “perceived organizational support”, and between worker and supervisor, termed “perceived supervisor support”, looking, for example, at their effect on workers’ engagement, retention, turnover, attitudes, and job satisfaction [12,15].

Perceived organizational support refers to the relationship between workers and their organizations [17]. Workers who perceive their organization as supportive of them and caring for their wellbeing will feel committed and obligated to help the organization achieve its goals [18]. Perceived supervisor support refers to the degree to which workers perceive their supervisors as valuing their contributions to the organization and as caring about their wellbeing. Perceived supervisor support was found to shape workers’ attitudes towards their work and their work effort [19], and was found to be associated with greater job involvement [20].

Although most studies dealing with organizational factors and workers’ performance have focused on formal organizational elements, researchers stress the importance of the interplay between formal and informal organizational networks as influencing workers’ job performance through various patterns, such as exchanging information and accessing resources [21,22]. Formal organization networks refer to formal connections between figures in an organization, such as between managers and their employees or between supervisors and supervisees, which represent formal workflow and authority relations [21]. On the other hand, informal organization networks refer to various forms of informal advice and information flow that connect workers within the organization, allow them to access information [23], and influence their job performance [24]. 

One important informal organization network is the relationship among colleagues. Perceived colleague support refers to the degree to which employees perceive that their co-workers respect them, care about them, and respect their contributions [25]. Perceived colleague support has been found to positively impact work attitude [26], organizational commitment [27], perceived quality of care delivery [2], and perceived preparedness to deliver emotional interventions [28].

Another informal factor, the perceived preparedness of colleagues, refers to the degree to which one perceives that his/her colleagues are prepared to conduct a program or an intervention. In a study on healthcare workers, perceived colleague preparedness was found to be associated with healthcare workers’ perceived preparedness to intervene in case of disasters [29]. Moreover, in a study of therapists who were trained in a new intervention, those who worked with colleagues who underwent the same training and used the same type of intervention were more likely to continue using the new intervention and treat more clients than those who worked with colleagues not trained or prepared to deliver the same intervention [30].

The current study builds on SET and on Soda and Zaheer’s (2012) [21] notion of the interplay between formal and informal networks within organizations, and considers both formal and informal relationships within the organization as influencing PHNs’ perceived preparedness to carry out three elements of a PPD screening program: screening, intervention, and referral [21]. PHNs’ perceived preparedness to carry out a PPD screening program is crucial for its success. However, the role of organizational factors in workers’ perceived preparedness to carry out a PPD screening program has yet to be examined, even though the impact of these factors on workers’ attitudes and job performance is significant. Therefore, the aim of this study was to examine the associations between perceived organizational support, supervisor’s support (formal organizational relationships), colleagues’ support and perceived colleagues’ preparedness (informal organizational relationships), and PHNs’ perceived preparedness to screen postpartum women for PPD symptoms, conduct an initial intervention, and refer women when deemed necessary.

## 2. Materials and Methods

### 2.1. Study Design and Participants

In this cross-sectional study, the sampling frame used was PHNs employed at the Ministry of Health’s 389 MCHCs in Israel. All these PHNs (N = 862) were eligible to participate in the study. Based on Barrett (2007), a sample size of at least 200 is adequate for structural equation modeling analyses (a path analysis among them) [31]. Taking into account a 53% response rate in an email survey [32], the sample size required was 380. Therefore, one nurse from each MCHC was randomly selected, reaching a total of 389 PHNs selected for the study. Of these PHNs, 39 did not have a correct email address, so they were substituted by the next nurse on their clinic’s roster. Eventually, 219 PHNs completed the survey (56.3% response rate) (Figure 1). In order to guarantee confidentiality, only the research assistant, who was not an employee of the Ministry of Health, knew which nurses were selected for the study.

### 2.2. Procedure

Prior to conducting the study, the Israeli Ministry of Health’s Helsinki Committee approval was obtained, and the head of the Ministry’s Public Health Nursing Service sent emails to all PHNs (n = 862) informing them of the planned study. A separate email was sent to each of those selected (n = 389), that included an explanation of the study’s purpose, a promise of confidentiality, the study questionnaire, an option to decline participation, and the fact that answering the survey indicated consent to participate (Figure 1). Participants were asked to return the survey to the research assistant by email or fax. The study was conducted between February and August 2016.

### 2.3. Measures

The Perceived Preparedness and Perceived Colleague Preparedness to Screen, Intervene and Refer questionnaires were designed especially for the study, based on questions adapted from the literature on perceived preparedness [33,34]. Face validity of these questionnaires was assessed through consultation with a group of five experts in PPD, nursing, and research design. Moreover, after conducting a pilot study to evaluate the survey questionnaire, adaptations were made based on comments received and after reaching a consensus among the experts.

Perceived preparedness was assessed by three items on a 4-point Likert-type scale, ranging from 1 (strongly disagree) to 4 (strongly agree), with a higher score indicating higher perceived preparedness. The items were: “I feel prepared to screen women for PPD symptoms”; “I feel prepared to provide supportive counseling and advice to women with PPD symptoms”; “I feel prepared to refer women with PPD symptoms to treatment”. Each item was considered a dependent variable and treated as pseudocontinuous, based on the distribution of the responses.

Perceived colleagues’ preparedness to screen, intervene and refer was assessed by three questions based on the “perceived preparedness” questions above on a 3-point Likert-type scale, ranging from 1 (disagree) to 3 (strongly agree). Each question was changed from focusing on the respondent’s perceived preparedness to the respondent’s perceived colleagues’ preparedness. These three questions were treated as a single scale, with a higher score indicating a higher level of perceived colleagues’ preparedness. Internal consistency was Cronbach alpha = 0.85.

Perceived organizational support was measured using the short Survey of Perceived Organizational Support [35], which includes eight items assessing workers’ perceived organizational support. Respondents rated the degree to which they agree with eight statements on a four-point Likert-type scale ranging from 1 (strongly disagree) to 4 (strongly agree), with a higher score indicating a higher level of perceived organizational support. The original scale consisted of a seven-point Likert-type scale and was shortened to a four-point scale to be consistent with other scales used in the study. Internal consistency was Cronbach alpha = 0.92.

Perceived supervisor’s support was measured by the Supervisory Supportiveness Scale used by Tsui, et al. (1997) [36]. One question, which assessed supervisors’ support specifically regarding advice on PPD issues, was added by the research team. This eight-question four-point Likert-Type scale, ranging from 1 (strongly disagree) to 4 (strongly agree), describes the extent to which workers perceive their supervisors as supportive of them and approachable, with a higher score indicating higher perceived support. The original scale consisted of a seven-point Likert-type scale and was shortened to a four-point scale to be consistent with other scales used in the study. Internal consistency was Cronbach alpha = 0.88

Perceived colleagues’ support was assessed using the Social Support Survey developed by House (1981) [37]. The survey includes nine statements referring to perceived colleagues’ support, in which respondents were asked to rate to what extent their colleagues performed the activity described. Possible responses were given on a four-point Likert-type scale, ranging from 1 (“to a very little extent”) to 4 (“to a very high extent”), with a higher score representing a higher level of perceived support. Internal consistency was Cronbach alpha = 0.96.

Background characteristics included questions regarding age, education (academic degree/no academic degree), nationality (Jewish/Arab), and professional experience as a nurse in MCHC.

### 2.4. Data Analysis

Based on the theoretical background presented above, the analytical procedure included a path analysis model to show associations between the independent variables (perceived organizational support, supervisor’s support, colleagues’ support, and perceived colleagues’ preparedness), and the three dependent variables (perceived preparedness to screen, intervene, and refer). Because professional experience as a nurse in MCHC was highly correlated with age (r = 0.78), only age was entered into the path analysis, together with education and nationality, as background variables. The Mplus V.8 statistical package was used to test the models [38] within a measurement invariance test framework, i.e., a comparison between groups within a multiple-group analysis framework. The Maximum Likelihood Rescaled estimator (MLR) was used, as it is robust for cases in which normal distribution cannot be assumed. Because this path analysis model resembles a regression model with seven independent variables (three background variables and four independent variables) and seven percent *R*^2^, the overall power of the model was 0.87, taking into account 219 cases included in the analysis. Values of *p* < 0.05 were considered statistically significant.

## 3. Results

Table 1 shows the background characteristics of nurses who participated in the study. Participants’ ages ranged from 26 to 64 years (M = 45.5; SD = 8.99). About half of them reported that they were of Jewish nationality and half of Arab nationality. 147 (67.1%) had academic degrees. The mean number of years of professional experience as a nurse in MCHC was 17, and length of professional experience as a nurse in MCHC ranged from 6 months to 37 years.

Table 2 presents nurses’ personal perceived preparedness to screen, intervene and refer in cases of suspected PPD, and organizational factors. The average perceived preparedness to screen for PPD symptoms was 3.58 (range 1–4), the average perceived preparedness to intervene (i.e., to provide supportive counseling and advice to women with PPD symptoms) was 3.20 (range 1–4), and the average perceived preparedness to refer women with PPD symptoms to treatment was 3.58 (range 1–4). The average score of the PHNs’ perceived degree of colleagues’ preparedness was 2.08 (SD = 0.64), supervisor’s support was 3.53 (SD = 0.49), organizational support was 2.89 (SD = 0.70), and colleagues’ support was 3.26 (SD = 0.67).

Table 3 presents the standardized estimation results of the path analysis between perceived colleagues’ preparedness, perceived supervisor’s support, perceived organizational support, and perceived colleagues’ support, and the three perceived preparedness constructs (perceived preparedness to screen, intervene and refer). The table is built according to the path analysis format, in such a way that the background variables are exogenous to the model, associated with perceived colleagues’ preparedness, perceived supervisor’s support, perceived organizational support, and perceived colleagues’ support, which are associated with the three perceived preparedness variables [38].

The path analysis model is displayed in Figure 2. The model fit is above the threshold, CFI = 0.988, TLI = 0.943, RMSEA = 0.042, Chi-Square = 12.48, df = 9, *p* = 0.190 [39].

### Correlates of Perceived Preparedness to Screen, Intervene and Refer

Age was significantly associated with the three perceived preparedness constructs, with older nurses having a higher level of perceived preparedness to screen (β = 0.22, *p* < 0.010), intervene (β = 0.27, *p* < 0.001) and refer to treatment (β = 0.20, *p* < 0.050). Education was significantly associated with perceived preparedness to screen; those with a higher level of education perceived themselves as better prepared to screen (β = 0.20, *p* < 0.011). Perceived colleagues’ preparedness was also significantly associated with the three perceived preparedness constructs; the higher the degree to which the nurses perceived their colleagues as prepared, the more likely they were to perceive themselves as being prepared to screen (β = 0.21, *p* < 0.010), intervene (β = 0.23, *p* < 0.010) and refer (β = 0.32, *p* < 0.010). We also found a statistically significant association between perceived supervisor’s support and perceived preparedness for PPD screening. Perceived supervisor’s support was positively associated with perceived preparedness to screen (β = 0.18, *p* < 0.050). In addition, a significant association was found between perceived organizational support and perceived preparedness to intervene. Perceived organizational support was positively associated with perceived preparedness to intervene (β = 0.17, *p* < 0.050). In an alternative path framework, professional experience as a nurse in MCHC, rather than age, was tested as an effect on the perceived preparedness outcomes, and similar but weaker results were found; thus, the current model was reported.

## 4. Discussion

This study is, to our knowledge, the first to examine the influence of organizational factors on PHNs’ perceived preparedness to screen, intervene and refer women with suspected PPD symptoms. The findings indicate unique associations between formal and informal organizational factors and PHNs’ perceived preparedness to conduct a PPD screening program. Regarding formal organizational relationships, the more that PHNs perceived their supervisors as supportive, the more likely they were to feel prepared to screen; and the more they perceived their organizations as supportive, the more likely they were to perceive themselves as prepared to offer supportive counseling and advice (i.e., to intervene). Similar findings were reported in a previous study conducted in primary care settings which reported that organizational factors contribute to the success or failure of alcohol screening and brief intervention implementation [13]. These findings can be explained in the context of the Social Exchange Theory [15] and studies on employee engagement [40], which indicate that the more workers feel that their organization is supportive and recognizes the importance of their work, the more likely it is that they will be more engaged in their work and perform better at their jobs. Therefore, it seems reasonable that, because providing supportive counseling requires skills that are more complex in the emotional domain than those required for screening or referral for treatment and are not generally acquired in nurses’ training [41], PHNs would want to feel a greater sense of support from their organizations, and not merely support from their supervisors, in order to carry out this more complex intervention. Moreover, it seems reasonable that PHNs would want to feel that their organizations are supportive of them in order to conduct an intervention for which they feel less prepared. In addition, because the focus of the MCHC PHNs’ training was on the screening aspect of the program [4], it is likely that both the supervisors and the PHNs themselves perceived the PPD program as focused mainly on screening, and that the supervisor’s support for this aspect was perceived as important for conducting the screening component of the PPD program.

Regarding informal organizational relationships, the current study’s finding that perceived colleagues’ support does not impact PHNs’ perceived preparedness to conduct the program is supported by other studies which found that, in contrast to perceived organizational and supervisor’s support, this type of support had the weakest impact on workers’ organizational commitment [27] and job engagement [42]. Because perceived preparedness is positively associated with job engagement and workers’ organizational commitment [43,44] it seems reasonable that perceived organizational and supervisor’s support have a greater impact on PHNs’ perceived preparedness to conduct an intervention than colleagues’ support.

Further, the greater the degree to which the PHNs perceived their colleagues’ preparedness to conduct the PPD screening program, the more likely they were to perceive themselves as prepared to screen, intervene, and refer to treatment when necessary. Similarly, other researchers have found that perceived colleagues’ preparedness was associated with workers’ own perceived preparedness to intervene in disasters [29] and to screen for intimate partner violence [14], and that subjects’ comparison to others was found to be associated with self-confidence and self-performance evaluation [45]. The fact that perceived colleagues’ preparedness had the greatest impact on perceived preparedness and was associated with all three perceived preparedness aspects is supported by the notion of collective identity, derived from social identity theory, which states that individuals value and relate to the group of which they are members, even if they do not feel close to group members [46].

Moreover, older PHNs perceived themselves as better prepared to screen, intervene, and refer, while the higher educational level was associated with a greater degree of perceived preparedness to screen, but not to intervene or to refer. This could be due to the fact that in the PHNs’ advanced nursing courses, the content focuses more on the screening aspect of the program. Older age could reflect more experience, and experience is associated with higher perceived preparedness to conduct interventions [47]. It could also reflect more positive attitudes towards carrying out the program, as being older is associated with more positive attitudes towards work tasks [48], and more positive attitudes towards work tasks were found to be associated with higher perceived preparedness to conduct a PPD screening program [8].

The findings of this study should be considered in light of certain limitations. One limitation pertains to the subjectivity of the constructs under examination. Self-assessed organizational, supervisor’s, and colleagues’ support and perceived colleagues’ preparedness may not reflect the actual support that PHNs received, due to social desirability, which may have caused participants to answer questions more favorably [49]. Additionally, such self-report data can introduce recall bias, and both of these facts may have skewed the results. However, because subjective perceptions are a crucial element in carrying out behavior [50], they are important in the context of this study as well. In addition, the sample of PHNs who participated in the study is representative of only half of Israeli MCHC PHNs, as the other MCHCs are operated by Health Maintenance Organizations and not by the Ministry of Health. Moreover, the sample size is smaller than recommended by power analyses. Although these points pose a limitation to the generalizability of the study’s findings, this study is, nevertheless, the first to offer a glance at the relationship between organizational factors and perceived preparedness to conduct a screening program for PPD, and as such offers valuable insights.

### Implications for Nursing Practice

This study highlights important organizational factors to consider when implementing a PPD screening program within various organizations and services. Although PHNs moderately to strongly agreed that their colleagues, supervisors, and organization supported them, there is a need to improve their perceived support for several reasons. First, levels of this support may be lower than reported, due to social desirability [49]; and second, PHNs’ level of perceived preparedness to intervene was relatively lower than their level of perceived preparedness to screen and refer, and therefore preparedness to intervene should be especially improved. If these types of support contribute to PHNs’ perceived preparedness, then they should be enhanced.

Perceived organizational support is important to workers [18], especially when implementing a task that requires complex skills for which workers were not originally trained. Thus, it is important for managers to meet front-line nurses and express appreciation and support for them before and during the implementation of a screening program such as the one described. Direct supervisors’ support is also important and should be provided with respect to all aspects of the screening program. If the supervisors do not have the skills to provide a certain type of support, e.g., support for the intervention aspect, then they should provide nurses with an alternative supportive figure, such as a designated nurse with expertise in intervention in cases of PPD, who could fill this supportive need gap.

Because perceived colleagues’ preparedness was the most significant factor associated with perceived preparedness for all three aspects of the program, services that implement a PPD screening program should encourage peer-group continuing learning meetings. At meetings of this type, nurses could share knowledge and experience regarding aspects of the program and discuss barriers and facilitators towards its implementation. Such learning groups can foster professional development [51].

## 5. Conclusions

Formal and informal organizational relationships play an important role in the perceived preparedness of PHNs to implement a PPD screening program for the early identification of symptoms of PPD, as well as to conduct interventions and refer women for treatment when necessary. Organizations that decide to implement such programs should support their health providers (e.g., nurses) in all aspects of the implementation process, provide regular guidance, training, and support, and enhance peer-group continued learning. From a health policy point of view, the development of policies which take into account organizational factors that may impact PHNs’ perceived preparedness to implement a screening program can ensure the appropriate implementation of such programs, and eventually reduce PPD.

Regarding future research, conducting a mixed methods study to examine PHNs’ perceived preparedness to carry out a screening program is recommended, in order to be able to triangulate data from multiple stakeholders across the MCHC system (physicians, nurses, clinic administrators, support staff, etc.) regarding the perceived preparedness of PHNs to screen, intervene and refer. Additionally, differences in the implementation of screening programs in other areas (e.g., substance abuse) were found to be associated with the location of the program (urban vs. rural) [52]. Moreover, healthcare providers (e.g., OB/GYNs, nurses, GPs) vary in the amount of time spent and the type of relationship with patients. For instance, nurses tend to spend more time with patients and form more intimate relationships with them, as compared with physicians [52]. Therefore, perceived preparedness to implement a screening program in various cultures and settings (e.g., rural and urban), while comparing various healthcare providers (e.g., OB/GYNs, nurses, GPs) and forms of screening implementation (e.g., filling out an electronic screening measure vs. face-to-face screening) could shed light on differences in perceived preparedness to screen, intervene and refer in cases of PPD, as well as on differences in factors associated with such preparedness. In addition, as healthcare systems are increasingly implementing the use of new and innovative technologies (e.g., artificial intelligence), which could assist healthcare staff in their work, especially when dealing with large numbers of clients [53,54], and which could be used for PPD screening [55], examining the perceived preparedness of healthcare providers (e.g., PHNs) to use such techniques in the PPD screening process is of importance. These studies could contribute to gaining a more in-depth understanding of ways to enhance PHNs’ preparedness to implement a PPD screening program, and ultimately to achieving the most optimal program.

## Figures and Tables

**Figure 1 ijerph-19-16717-f001:**
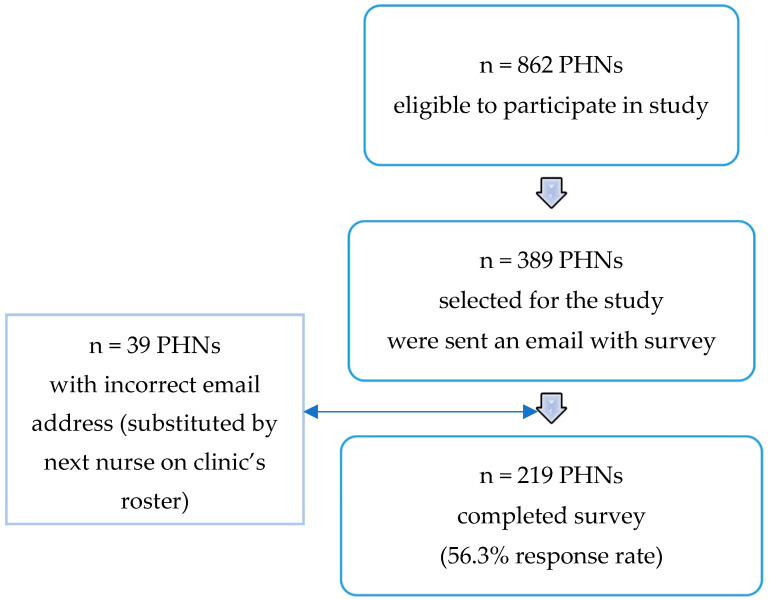
Research procedure flowchart.

**Figure 2 ijerph-19-16717-f002:**
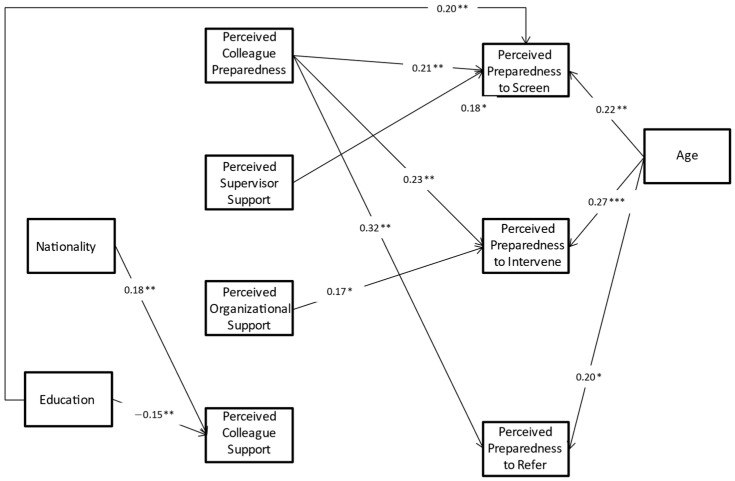
Path analysis graphic model results, standardized coefficients. * Only significant (*p* < 0.05) estimates are shown. * *p* < 0.05, ** *p* < 0.01, *** *p* < 0.001.

**Table 1 ijerph-19-16717-t001:** Background characteristics of nurses who participated in the study (N = 219).

Characteristics	Mean (SD), Range/N (%)
Age	45.53 (8.9), 26–64
Nationality	
Jewish	110 (50%)
Arab	109 (50%)
Education	
Academic degree	147 (67.1%)
No academic degree	64 (29.2%)
Professional experience as a nurse in MCHC *	17 (9.1), 0.5–37

* Maternal and Child Healthcare Center.

**Table 2 ijerph-19-16717-t002:** Nurses’ personal perceived preparedness to screen, intervene and refer in cases of suspected postpartum depression, and organizational factors (N = 219).

Characteristics	Mean (SD), Range/N (%)
Personal perceived preparedness	
PHNs’ * perceived preparedness to screen women for PPD ° symptoms	3.58 (0.40), 1–4
PHNs’ average perceived preparedness to intervene (to provide supportive counseling and advice to women with PPD symptoms)	3.20 (0.50), 1–4
PHNs’ perceived preparedness to refer women with PPD symptoms to treatment	3.58, (0.40), 1–4
Organizational factors	
Colleagues’ perceived preparedness	2.08 (0.64), 1–3
Perceived supervisor’s support regarding advice on PPD issues	3.53 (0.49), 1–4
Perceived organizational commitment	2.89 (0.70), 1–4
Perceived colleagues’ support	3.26 (0.67), 1–4

* Public health nurses, ° Postpartum depression.

**Table 3 ijerph-19-16717-t003:** Path analysis results, standardized coefficients (N = 219).

	Perceived Colleagues’ Preparedness	Perceived Supervisor’s Support	Perceived Organizational Support	Perceived Colleagues’ Support	Perceived Preparedness to Screen	Perceived Preparedness to Intervene	Perceived Preparedness to Refer
Age	-	-	-	-	0.22 ** (0.07)	0.27 *** (0.06)	0.20 * (0.07)
Nationality	-	-	-	0.18 ** (0.06)	0.09 (0.06)	−0.07 (0.06)	−0.08 (0.06)
Education	-	-	-	−0.17 ** (0.06)	0.20** (0.06)	0.07 (0.07)	0.09 (0.06)
Perceived colleagues’ preparedness	-				0.21 ** (0.06)	0.23 ** (0.07)	0.32 ** (0.05)
Perceived supervisor’s support	0.14 *	-			0.18 * (0.07)	0.07 (0.08)	0.15 (0.08)
Perceived organizational commitment	0.22 **	0.38 ***	-		0.10 (0.07)	0.17* (0.08)	0.04 (0.08)
Perceived colleagues’ support	0.15 *	0.41 ***	0.36 ***	-	0.04 (0.07)	0.12 (0.08)	0.05 (0.07)
Perceived preparedness to screen					-		
Perceived preparedness to intervene					0.44 ***	-	
Perceived preparedness to refer					0.47 ***	0.28 ***	-
R^2^	0.00 (0.00)	0.00 (0.00)	0.00 (0.00)	0.06 * (0.03)	0.20 *** (0.05)	0.23 *** (0.05)	0.20 *** (0.05)

* *p* < 0.05, ** *p* < 0.01, *** *p* < 0.001; shaded cells for model correlations; standard errors in parentheses.

## Data Availability

Anonymized data are available from the corresponding author upon reasonable request.

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
