# Peer review of "The Role of Organizational Factors in Nurses’ Perceived Preparedness to Screen, Intervene and Refer in Cases of Suspected Postpartum Depression"

_ijerph, 2022, doi:10.3390/ijerph192416717_

Round 1

Reviewer 1 Report

This study is a cross-sectional survey study of 219 public health nurses (PHNs) working in  Mother-and-Child Health Clinics in Israel. The cross-sectional survey dataset appears to have already been used to address a different set of questions relating to public health nurses “perceived preparedness to screen, intervene and refer women with suspected post-partum depression,” while this study using the same cross-sectional survey dataset to address the issue of institutional factors such as their “perceived organizational support, supervisor's support, colleagues' support, and colleagues’ preparedness.”

Indeed a PubMed search using the terms [“nurses” and post-partum depression” and “screening” and “organizational factors”] reveals no existing studies specifically on organizational factors as they relate to nurses perceived preparedness to screen, intervene and refer, supporting the authors conclusions that this is a novel study. However, it is also clear that this study uses the same dataset of cross-sectional survey answers obtained from 219 public health nurses working in Mother-and-Child Health Clinics in Israel (PMID: 31207447):

Bina R, Glasser S, Honovich M, Levinson D, Ferber Y. Nurses perceived preparedness to screen, intervene, and refer women with suspected postpartum depression. Midwifery. 2019 Sep;76:132-141. doi: 10.1016/j.midw.2019.05.009. Epub 2019 May 25. PMID: 31207447.

 So this appears to be the next piece of research stemming from this single cross-sectional study conducted between February and August of 2019. Obviously, there is a limitation to the self-report survey model as it can affect recall and introduce bias (PMID: 25383095), and in future it may prove more powerful to perform a mixed methods study in order to be able to triangulate the data from multiple stakeholders across the clinic system (physicians, nurses, clinic administrators, support staff, etc.) about the perceived preparedness of public health nurses to screen, intervene and refer. Furthermore, it is clear that the survey instrument questions although showing good internal consistency (via Chronbach alpha score) are not validated instruments, which would make the findings more substantial. It is unclear from the following statement whether the authors are suggesting that the entire survey was indeed validated: “Face validity of the entire survey was assessed through consultation with a group of five experts in PPD, nursing, and research design. Moreover, after conducting a pilot study to evaluate the survey questionnaire, adaptations were made based on comments received and after reaching a consensus among the experts.”

This reviewer’s understanding of path analysis is that it is best used to compare a priori different models to determine a best fit that explains the greatest degree of variance, but it appears that the authors have only presented one a priori model and failed to create any other a priori models for best-fit comparison. An explanation of why this is so would be appreciated. Furthermore, the statement “Since professional experience as a nurse in MCHC (was highly correlated with age (r = .78), only age was entered into the path analysis together with education and nationality as background variables.” Normally one could compare the effect on overall variance of the variables and factors stemming from outside the model (external effects including measurement error). It appears that the authors have just assumed that age, education, and nationality are the only external factors that affect the variance, but why not actually test for this by varying the model?

 Although there clearly has been a lot of work done on organizational factors affecting nurses performing primary care alcohol screening and brief intervention (PMID: 16047525)(with organizational factors such as: prior SBI experience, managed care organization stability, number of clinicians trained and the quality of the managed care organization (MCO) coordinator's work); intimate partner and sexual violence screening and intervention (PMID: 31087366), it does not appear that this has been done for post-partum depression, where most of the research to date has focused on implementation and education about the importance of screening for post-partum depression, development of survey instruments and analysis of the potential causes of post-partum depression (such as socio-demographics, mode of birth, prenatal attachment, breastfeeding, etc.)(PMID: 26694512). Urban versus rural location of the clinic location from existing published literature would also have been powerful as large difference have been observed for example for screening and intervention for intimate partner violence and sexual violence. Also examination of a post-partum screening prompts built into the electronic health record (HER) would also effect perceived willingness to screen, intervene and refer as this is technically also an organizational factor determined by the administration and thereby demanding an action from the practitioner (you cannot just leave it blank). Similarly, examining the perception of public health nurses to screen, intervene and refer compared to other healthcare providers would be of interest given that it is a well-known fact that nurses tend to spend much more time with patients and form more intimate bonds with them (PMID: 29649458). It would also have been interesting to see differences in screening, intervention and referral by provider type as has been done in several other studies (but not for past partum depression). 

Line 21-23:  For this sentence in the abstract please include the statement about also intervening: “The aim of the study was to examine organizational factors associated with public health nurses (PHNs) perceived preparedness to screen women for PPD, and refer women in cases of suspected PPD.”

 Line 244; please add a period at the end of this sentence: “Correlates of perceived preparedness to screen, intervene and refer”

Author Response

Dear reviewer,

Thank you so much for your helpful feedback, comments, and suggestions, which helped to improve the quality, readability, and impact of our manuscript. We hope that you will find this revised manuscript suitable for publication.

Reviewer 2 Report

I hope you find the following observations helpful:

The introduction needs to show more paper dimensions because most readers just read the abstract. Thus, please develop the abstract and briefly list the main contribution of this review paper.

Materials and methods: I found this section very important for the paper's readability. Methods should be described in detail. I think the research procedure could be more clearly described using a diagram highlighting its potential and limit.

Authors should consider more previous works (e.g., theoretical, conceptual, and empirical reviews) published in the literature. Authors should discuss the results and how they can be interpreted from the perspective of previously published studies:

1. Kumar, P., Savadatti, M. Virobot the Artificial Assistant Nurse for Health Monitoring, Telemedicine and Sterilization through the Internet. International Journal of Wireless and Microwave Technologies 2020, Vol.10, No.6, pp. 16-26. DOI: 10.5815/ijwmt.2020.06.03

2. Fedushko S., Michal Gregus ml., Ustyianovych T. Medical card data imputation and patient psychological and behavioral profile construction.  Procedia Computer Science. Vol. 160, 2019, pp. 354-361. https://doi.org/10.1016/j.procs.2019.11.080

I strongly recommend adding these works to the list of references.

The paper reviews a certain topic, but the title suggests that it is a description of some new method. It needs to be corrected.

Authors should discuss the results and how they can be interpreted from the perspective of previously published studies.

The diagrammatic presentation of the study research will be the most substantial section of this work. I suggest adding a visual presentation of obtained outcomes in the section Results.

I also suggest a grammar and spelling review. 

Author Response

(The authors gave the same response as above.)

Round 2

Reviewer 2 Report

 Accept in present form